# Direct 0-1 Loss Minimization and Margin Maximization with Boosting

**Shaodan Zhai, Tian Xia, Ming Tan and Shaojun Wang**
Kno.e.sis Center
Department of Computer Science and Engineering
Wright State University
{zhai.6,xia.7,tan.6,shaojun.wang}@wright.edu

## Abstract

We propose a boosting method, DirectBoost, a greedy coordinate descent algorithm that builds an ensemble classifier of weak classifiers through directly minimizing empirical classification error over labeled training examples; once the training classification error is reduced to a local coordinatewise minimum, Direct-Boost runs a greedy coordinate ascent algorithm that continuously adds weak classifiers to maximize any targeted arbitrarily defined margins until reaching a local coordinatewise maximum of the margins in a certain sense. Experimental results on a collection of machine-learning benchmark datasets show that DirectBoost gives better results than AdaBoost, LogitBoost, LPBoost with column generation and BrownBoost, and is noise tolerant when it maximizes an $n'$th order bottom sample margin.

## 1 Introduction

The classification problem in machine learning and data mining is to predict an unobserved discrete output value $y$ based on an observed input vector $x$. In the spirit of the model-free framework, it is always assumed that the relationship between the input vector and the output value is stochastic and described by a fixed but unknown probability distribution $p(X, Y)$ [7]. The goal is to learn a classifier, i.e., a mapping function $f(x)$ from $x$ to $y \in \mathcal{Y}$ such that the probability of the classification error is small. As it is well known, the optimal choice is the Bayes classifier [7]. However, since $p(X, Y)$ is unknown, we cannot learn the Bayes classifier directly. Instead, following Vapnik's general setting of the empirical risk minimization [7, 24], we focus on a more realistic goal: Given a set of training data $\mathcal{D} = \{(x_1, y_1), \cdots, (x_n, y_n)\}$ independently drawn from $p(X, Y)$, we consider finding $f(x)$ in a function class $\mathcal{H}$ that minimizes the empirical classification error,

$$\frac{1}{n} \sum_{i=1}^{n} \mathbf{1}(\hat{y}_i \neq y_i) \tag{1}$$

where $\hat{y}_i = \arg\max_{y \in \mathcal{Y}} y f(x_i)$, $\mathcal{Y} = \{-1, 1\}$ and $\mathbf{1}(\cdot)$ is an indicator function. Under certain conditions, direct empirical classification error minimization is consistent [24] and under low noise situations it has a fast convergence rate [15, 23]. However, due to the nonconvexity, nondifferentiability and discontinuity of the classification error function, the minimization of (1) is typically NP-hard for general linear models [13]. The common approach is to minimize a surrogate function which is usually a convex upper bound of the classification error function. The problem of minimizing the empirical surrogate loss turns out to be a convex programming problem with considerable computational advantages and learned classifiers remain consistent to Bayes classifier [1, 20, 28, 29], however clearly there is a mismatch between "desired" loss function used in inference and "training" loss function during the training process [16]. Moreover, it has been shown that all boosting algorithms based on convex functions are susceptible to random classification noise [14].

Boosting is a machine-learning method based on the idea of creating a single, highly accurate classifier by combining many weak and inaccurate "rules of thumb." A remarkably rich theory and a record of empirical success [18] have evolved around boosting, nevertheless it is still not clear how to best exploit what is known about how boosting operates, even for binary classification. In

this paper, we propose a boosting method for binary classification – DirectBoost – a greedy coordinate descent algorithm that directly minimizes classification error over labeled training examples to build an ensemble linear classifier of weak classifiers. Once the training error is reduced to a (local coordinatewise) minimum, DirectBoost runs a coordinate ascent algorithm that greedily adds weak classifiers by directly maximizing any targeted arbitrarily defined margins, it might escape the region of minimum training error in order to achieve a larger margin. The algorithm stops once a (local coordinatewise) maximum of the margins is reached. In the next section, we first present a coordinate descent algorithm that directly minimizes 0-1 loss over labeled training examples. We then describe coordinate ascent algorithms that aims to directly maximize any targeted arbitrarily defined margins right after we reach a (local coordinatewise) minimum of 0-1 loss. In Section 3, we show experimental results on a collection of machine-learning benchmark data sets for DirectBoost, AdaBoost [9], LogitBoost [11], LPBoost with column generation [6] and BrownBoost [10], and discuss our findings. Due to space limitation, the proofs of theorems, related works, technical details as well as conclusions and future works are given in the full version of this paper [27].

## 2  DirectBoost: Minimizing 0-1 Loss and Maximizing Margins

Let $\mathcal{H} = \{h_1, ..., h_l\}$ denote the set of all possible weak classifiers that can be produced by the weak learning algorithm, where a weak classifier $h_j \in \mathcal{H}$ is a mapping from an instance space $\mathcal{X}$ to $\mathcal{Y} = \{-1, 1\}$. The $h_j$s are not assumed to be linearly independent, and $\mathcal{H}$ is closed under negation, i.e., both $h$ and $-h$ belong to $\mathcal{H}$. We assume that the training set consists of examples with labels $\{(x_i, y_i)\}, i = 1, \cdots, n$, where $(x_i, y_i) \in \mathcal{X} \times \mathcal{Y}$ that are generated independently from $p(X, Y)$. We define $\mathcal{C}$ of $\mathcal{H}$ as the set of mappings that can be generated by taking a weighted average of classifiers from $\mathcal{H}$:

$$\mathcal{C} = \left\{ f : x \to \sum_{h \in \mathcal{H}} \alpha_h h(x) \mid \alpha_h \geq 0 \right\}, \tag{2}$$

The goal here is to find $f \in \mathcal{C}$ that minimizes the empirical classification error (1), and has good generalization performance.

### 2.1  Minimizing 0-1 Loss

Similar to AdaBoost, DirectBoost works by sequentially running an iterative greedy coordinate descent algorithm, each time directly minimizing *true* empirical classification error (1) instead of a weighted empirical classification error in AdaBoost. That is, for each iteration, only the parameter of a weak classifier that leads to the most significant true classification error reduction is updated, while the weights of all other weak classifiers are kept unchanged. The rationale is that the inference used to predict the label of a sample can be written as a linear function with a single parameter.

Consider the $t$th iteration, the ensemble classifier is

$$f_t(x) = \sum_{k=1}^{t} \alpha_k h_k(x) \tag{3}$$

where previous $t - 1$ weak classifiers $h_k(x)$ and corresponding weights $\alpha_k, k = 1, \cdots, t - 1$ have been selected and determined. The inference function for sample $x_i$ is defined as

$$F_t(x_i, y) = y f_t(x_i) = y \left( \sum_{k=1}^{t-1} \alpha_k h_k(x_i) \right) + \alpha_t y h_t(x_i) \tag{4}$$

Since $a(x_i) = \sum_{k=1}^{t-1} \alpha_k h_k(x_i)$ is constant and $h_k(x_i)$ is either +1 or -1 depending on sample $x_i$, we re-write the equation above as,

$$F_t(x_i, y) = y\, h_t(x_i) \alpha_t + y a(x_i) \tag{5}$$

Note that for each label $y$ of sample $x_i$, there is a linear function of $\alpha_t$ with the slope to be either +1 or -1 and intercept to be $ya(x_i)$. Given an input of $\alpha_t$, each example $x_i$ has two linear scoring functions, $F_t(x_i, +1)$ and $F_t(x_i, -1)$, $i = 1, \cdots, n$, one for the positive label $y = +1$ and one for the negative label $y = -1$. From these two linear scoring functions, the one with the higher score determines the predicted label $\hat{y}_i$ of the ensemble classifier $f_t(x_i)$. The intersection point $e_i$ of these two linear scoring functions is the critical point that the predicted label $\hat{y}_i$ switches its sign, the intersection point satisfies the condition that $F_t(x_i, +1) = F_t(x_i, -1) = 0$, i.e. $a(x_i) + \alpha_t h_t(x_i) = 0$, and can be computed as $e_i = -\frac{a(x_i)}{h_t(x_i)}, i = 1, \cdots, n$. These points divide $\alpha_t$ into (at most) $n + 1$ intervals, each interval has the value of a true classification error, thus the classification error is a stepwise

**Algorithm 1** Greedy coordinate descent algorithm that minimizes a 0-1 loss.

1: $\mathcal{D} = \{(x_i, y_i), i = 1, \cdots, n\}$
2: Sort $|a(x_i)|, i = 1, \cdots, n$ in an increasing order.
3: **for** a weak classifier $h_k \in \mathcal{H}$ **do**
4:     Visit each sample in the order that $|a(x_i)|$ is increasing.
5:         Compute the slope and intercept of $F(x_i, y_i) = y_i h_k(x_i)\alpha + y_i a(x_i)$.
6:         Let $\hat{e}_i = |a(x_i)|$.
7:         If (slope $> 0$ and intercept $< 0$), error update on the righthand side of $\hat{e}_i$ is -1.
8:         If (slope $< 0$ and intercept $> 0$), error update on the righthand side of $\hat{e}_i$ is +1.
9:     Incrementally calculate classification error on intervals of $\hat{e}_i s$.
10:     Get the interval that has minimum classification error.
11: **end for**
12: Pick the weak classifiers that lead to largest classification error reduction.
13: Among selected these weak classifiers, only update the weight of one weak classifier that gives the smallest exponential loss.
14: Repeat 2-13 until training error reaches minimum.

---

function of $\alpha_t$. The value of $e_i, i = 1, \cdots, n$ can be negative or positive, however since $\mathcal{H}$ is closed in negation, we only care about these that are positive.

The greedy coordinate descent algorithm that sequentially minimizes a 0-1 loss is described in Algorithm 1, lines 3-11 are the weak learning steps and the rest are boosting steps. Consider an example with 4 samples to illustrate this procedure. Suppose for a weak classifier, we have $F_t(x_i, y_i), i = 1, 2, 3, 4$ as shown in Figure 1. At $\alpha_t = 0$, samples $x_1$ and $x_2$ have negative margins, thus they are misclassified, the error rate is 50%. We incrementally update the classification error on intervals of $\hat{e}_i, i = 1, 2, 3, 4$: For $F_t(x_1, y_1)$, its slope is negative and its intercept is negative, sample $x_1$ always has a negative margin for $\alpha_t > 0$, thus there is no error update on the right-hand side of $\hat{e}_1$. For $F_t(x_2, y_2)$, its slope is positive and its intercept is negative, then when $\alpha_t$ is at the right side of $\hat{e}_2$, sample $x_2$ has positive margin and becomes correctly classified, so we update the error by -1, the error rate is reduced to 25%. For $F_t(x_3, y_3)$, its slope is negative and its intercept is positive, then when $\alpha_t$ is at the right side of $\hat{e}_3$, sample $x_3$ has a negative margin and becomes misclassified, so we update the error rate changes to 50% again. For $F_t(x_4, y_4)$, its slope is positive and its intercept is positive, sample $x_4$ always has positive margin for $\alpha_t > 0$, thus there is no error update on the right-hand side of $\hat{e}_4$. We finally have the minimum error rate of 25% on the interval of $[\hat{e}_2, \hat{e}_3]$.

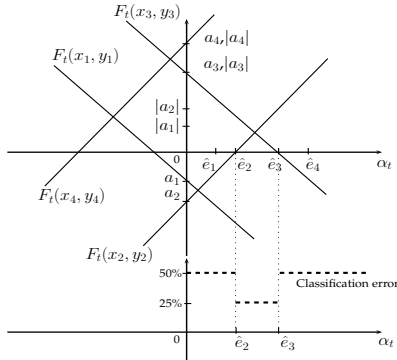

Figure 1: *An example of computing minimum 0-1 loss of a weak learner over 4 samples.*

We repeat this procedure until the training error reaches its minimum, which may be 0 in a data separable case. We then go to the next stage, explained below, that aims to maximize margins. A nice property of the above greedy coordinate descent algorithm is that the classification error is monotonically decreasing. Assume there are $M$ weak classifiers be considered, the computational complexity of Algorithm 1 in the training stage is $O(Mn)$ for each iteration.

For boosting, as long as the weaker learner is strong enough to achieve reasonably high accuracy, the data will be linearly separable and the minimum 0-1 loss is usually 0. As shown in Theorem 1, the region of zero 0-1 loss is a (convex) cone.

**Theorem 1** *The region of zero training error, if exists, is a cone, and it is not a set of isolated cones.*

Algorithm 1 is a heuristic procedure that minimizes 0-1 loss, it is not guaranteed to find the global minimum, it may trap to a coordinatewise local minimum [22] of 0-1 loss. Nevertheless, we switch to algorithms that directly maximize the margins we present below.

## 2.2 Maximizing Margins

The margins theory [17] provides an insightful analysis for the success of AdaBoost where the authors proved that the generalization error of any ensemble classifiers is bounded in terms of the

entire distribution of margins of training examples, as well as the number of training examples and the complexity of the base classifiers, and AdaBoost's dynamics has a strong tendency to increase the margins of training examples. Instead, we can prove that the generalization error of any ensemble classifier is bounded in terms of the average margin of bottom $n'$ samples or $n'$th order margin of training examples, as well as the number of training examples and the complexity of the base classifiers. This view motivates us to propose a coordinate ascent algorithm to directly maximize several types of margins just right after the training error reaches a (local coordinatewise) minimum.

The margin of a labeled example $(x_i, y_i)$ with respect to an ensemble classifier $f_t(x) = \sum_{k=1}^{t} \alpha_k h_k(x_i)$ is defined to be

$$m_i = \frac{y_i \sum_{k=1}^{t} \alpha_k h_k(x_i)}{\sum_{k=1}^{t} \alpha_k} \tag{6}$$

This is a real number between -1 and +1 that intuitively measures the confidence of the classifier in its prediction on the $i$th example. It is equal to the weighted fraction of base classifiers voting for the correct label minus the weighted fraction voting for the incorrect label [17].

We denote the minimum margin, the average margin, and median margin over the training examples as $g_{\min} = \min_{i \in \{1, \cdots, n\}} m_i$, $g_{\text{average}} = \frac{1}{n} \sum_{i=1}^{n} m_i$, and $g_{\text{median}} = \text{median}\{m_i, i = 1, \cdots, n\}$. Furthermore, we can sort the margins over all training examples in an increasing order, and consider $n'$ worst training examples $n' \leq n$ that have smaller margins, and compute the average margin over those $n'$ training examples. We call this the average margin of the bottom $n'$ samples, and denote it as $g_{\text{average } n'} = \frac{1}{n'} \sum_{i \in B_{n'}} m_i$, where $B_{n'}$ denotes the set of $n'$ samples having the smallest margins.

The margin maximization method described below is a greedy coordinate ascent algorithm that adds a weak classifier achieving maximum margin. It allows us to continuously maximize the margin while keeping the training error at a minimum by running the greedy coordinate descent algorithm presented in the previous section. The margin $m_i$ is a *linear fractional function* of $\underline{\alpha}$, and it is quasiconvex, and quasiconcave, i.e., *quasilinear* [2, 5]. Theorem 2 shows that the average margin of bottom $n'$ examples is quasiconcave in the region of the zero training error.

**Theorem 2** *Denote the average margin of bottom $n'$ samples as*

$$g_{\text{average } n'}(\underline{\alpha}) = \sum_{i \in \{B_{n'} | \underline{\alpha}\}} \frac{y_i \sum_{k=1}^{t} \alpha_k h_k(x_i)}{\sum_{k=1}^{t} \alpha_k}$$

*where $\{B_{n'} | \underline{\alpha}\}$ denotes the set of $n'$ samples whose margins are at the bottom for fixed $\underline{\alpha}$. Then $g_{\text{average } n'}(\underline{\alpha})$ in the region of zero training error is quasiconcave.*

We denote $a_i = \sum_{k=1}^{t-1} y_i \alpha_k h_k(x_i)$, $b_{i,t} = y_i h_t(x_i) \in \{-1, +1\}$ and $c = \sum_{k=1}^{t-1} \alpha_k$, then the margin on the $i$th example $(x_i, y_i)$ can be rewritten as,

$$m_i = \frac{a_i + b_{i,t} \alpha_t}{c + \alpha_t} \tag{7}$$

The derivative of the margin on $i$th example with respect to $\alpha_t$ is calculated as,

$$\frac{\partial m_i}{\partial \alpha_t} = \frac{b_{i,t} c - a_i}{(c + \alpha_t)^2} \tag{8}$$

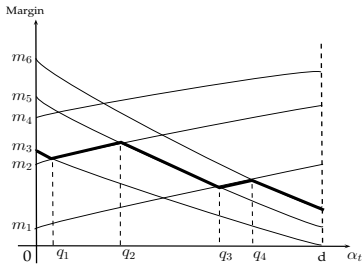

Since $c \geq a_i$, depending on the sign of $b_{i,t}$, the derivative of the margin on the $i$th sample $(x_i, y_i)$ is either positive or negative, which is irrelevant to the value of $\alpha_t$. This is also true for the second derivative of the margin. Therefore, the margin on the $i$th example $(x_i, y_i)$ with respect to $\alpha_t$ is either concave when it is monotonically increasing or convex when it is monotonically decreasing. See Figure 2 for a simple illustration.

Figure 2: *Margin curves of six examples. At points $q_1, q_2, q_3$ and $q_4$, the median example is changed. At points $q_2$ and $q_4$, the set of bottom $n' = 3$ examples are changed.*

Consider a greedy coordinate ascent algorithm that maximizes the average margin $g_{\text{average}}$ over all training examples. The derivative of $g_{\text{average}}$ can be written as,

$$\frac{\partial g_{\text{average}}}{\partial \alpha_t} = \frac{\sum_{i=1}^{n} b_{i,t} c - \sum_{i=1}^{n} a_i}{(c + \alpha_t)^2} \tag{9}$$

**Algorithm 2** Greedy coordinate ascent algorithm that maximizes the average margin of bottom $n'$ examples.

1: Input: $a_{i=1,\cdots,n}$ and $c$ from previous round.
2: Sort $a_{i=1,\cdots,n}$ in an increasing order. $B_{n'} \leftarrow \{n' \text{ samples having the smallest } a_i \text{ at } \alpha_t = 0\}$.
3: **for** a weak classifier **do**
4:     Determine the lowest sample whose margin is decreasing and determine $d$.
5:     Compute $D_{n'} \leftarrow \sum_{i \in B_{n'}} (b_{i,t}c - a_i)$.
6:     $j \leftarrow 0, q_j \leftarrow 0$.
7:     Compute the intersection $q_{j+1}$ of the $j+1$th highest increasing margin in $B_{n'}$ and the $j+1$th smallest decreasing margin in $B_{n'}^c$ (the complement of the set $B_{n'}$).
8:     **if** $q_{j+1} < d$ and $D_{n'} > 0$ **then**
9:         Incrementally update $B_{n'}$, $B_{n'}^c$ and $D_{n'}$ at $\alpha_t = q_{j+1}$; $j \leftarrow j+1$.
10:        Go back to Line 7.
11:     **else**
12:        if $D_{n'} > 0$ then $q^* \leftarrow d$; otherwise $q^* \leftarrow q_j$.
13:        Compute the average margin of the bottom $n'$ examples at $q^*$.
14:     **end if**
15: **end for**
16: Pick the weak classifier with the largest increment of the average margin of bottom $n'$ examples with weight being $q^*$.
17: Repeat 2-16 until no increment in average margin of bottom $n'$ examples.

Therefore, the maximum average margin can only happen at two ends of the interval. As shown in Figure 2, the maximum average margin is either at the origin or at point $d$, which depends on the sign of the derivative in (9). If it is positive, the average margin is monotonically increasing, we set $\alpha_t = d - \epsilon$, otherwise we set $\alpha_t = 0$. The greedy coordinate ascent algorithm found by: looking at all weak classifiers in $\mathcal{H}$, if the nominator in (9) is positive, we let its weight $\epsilon$ close to the right value on the interval where the training error is minimum, and compute the value of the average margin. We add the weak classifier which has the largest average margin increment. We iterate this procedure until convergence. Its convergence is given by Theorem 3 shown below.

**Theorem 3** *When constrained to the region of zero training error, the greedy coordinate ascent algorithm that maximizes the average margin over all examples converges to an optimal solution.*

Now consider a greedy coordinate ascent algorithm maximizing the average margin of bottom $n'$ training examples, $g_{\text{average n}'}$. Apparently maximizing the minimum margin is a special case by choosing $n' = 1$. Figure 2 is a simple illustration with six training examples. Our aim is to maximize the average margin of the bottom 3 examples. The interval $[0, d]$ of $\alpha_t$ indicates an interval where the training error is zero. On the point of $d$, the sample margin $m_3$ alters from positive to negative, which causes the training error jump from 0 to 1/6. As shown in Figure 2, the margin of each of six training examples is either monotonically increasing or decreasing.

If we know a fixed set of bottom $n'$ training examples having smaller margins for an interval of $\alpha_t$ with a minimum training error, it is straightforward to compute the derivative of the average margin of bottom $n'$ training examples as

$$\frac{\partial g_{\text{average n}'}}{\partial \alpha_t} = \frac{\sum_{i \in B_{n'}} b_{i,t}c - \sum_{i \in B_{n'}} a_i}{(c + \alpha_t)^2} \tag{10}$$

Again $g_{\text{average n}'}$ is a monotonic function of $\alpha_t$, depending on the sign of the derivative in (10), it is maximized either on the left side or on the right side of the interval.

In general, the set of bottom $n'$ training examples for an interval of $\alpha_t$ with a minimum training error varies over $\alpha_t$, it is required to precisely search for any snapshot of bottom $n'$ examples with a different value of $\alpha$.

To address this, we first examine when the margins of two examples intersect. Consider the $i$th example $(x_i, y_i)$ with margin $m_i = \frac{a_i + b_{i,t}\alpha_t}{c + \alpha_t}$ and the $j$th example $(x_j, y_j)$ with margin $m_j = \frac{a_j + b_{j,t}\alpha_t}{c + \alpha_t}$. Notice $b_i, b_j$ is either -1 or +1. Assume $b_i = b_j$, then because $m_i \neq m_j$ (since $a_i \neq a_j$), the margins of example $i$ and example $j$ never intersect; assume $b_i \neq b_j$, then because $m_i = m_j$

at $\alpha_t = \frac{|a_i - a_j|}{2}$, the margins of example $i$ and example $j$ might intersect with each other if $\frac{|a_i - a_j|}{2}$ belongs to the interval of $\alpha_t$ with the minimum training error. In summary, given any two samples, we can decide whether they intersect by checking whether $b$ terms have the same sign, if not, they do intersect, and we can determine the intersection point.

The greedy coordinate ascent algorithm that sequentially maximizes the average margin of bottom $n'$ examples is described in Algorithm 2, lines 3-15 are the weak learning steps and the rest are boosting steps. At line 5 we compute $D_{n'}$ which can be used to check the sign of the derivative in (10). Since the function of the average margin of bottom $n'$ examples is quasiconcave, we can determine the optimal point $q^*$ by $D_{n'}$, and only need to compute the margin value at $q^*$. We add the weak learner, which has the largest increment of the average margin over bottom $n'$ examples, into the ensembled classifier. This procedure terminates if there is no increment in the average margin of bottom $n'$ examples over the considered weak classifiers. If $M$ weak learners are considered, the computational complexity of Algorithm 2 in the training stage is $O\left(\max(n \log n, Mn')\right)$ for each iteration. The convergence analysis of Algorithm 2 is given by Theorem 4.

**Theorem 4** *When constrained to the region of zero training error, the greedy coordinate ascent algorithm that maximizes average margin of bottom $n'$ samples converges to a coordinatewise maximum solution, but it is not guaranteed to converge to an optimal solution due to the non-smoothness of the average margin of bottom $n'$ samples.*

$\epsilon$-**relaxation**: Unfortunately, there is a fundamental difficulty in the greedy coordinate ascent algorithm that maximizes the average margin of bottom $n'$ samples: It gets stuck at a corner, from which it is impossible to make progress along any coordinate direction. We propose an $\epsilon$-relaxation method to overcome this difficulty. This method was first proposed by [3] for the assignment problem, and was extended to the linear cost network flow problem and strictly convex costs and linear constraints [4, 21]. The main idea is to allow a single coordinate to change even if this worsens the margin function. When a coordinate is changed, it is set to $\epsilon$ plus or $\epsilon$ minus the value that maximizes the margin function along that coordinate, where $\epsilon$ is a positive number.

We can design a similar greedy coordinate ascent algorithm to directly maximize the bottom $n'$th sample margin by only making a slight modification to Algorithm 2: for a weak classifier, we choose the intersection point that led to the largest increasing of the bottom $n'$th margin. When combined with $\epsilon$-relaxation, this algorithm will eventually approach a small neighbourhood of a local optimal solution that maximizes the bottom $n'$th sample margin. As shown in Figure 2, bottom $n'$th margin is a multimodal function, this algorithm with $\epsilon$-relaxation is very sensitive to the choice of $n'$, and it usually gets stuck in a bad coordinatewise point without using $\epsilon$-relaxation. However, an impressive advantage is that this method is tolerant to noise, which will be shown in Section 3.

## 3 Experimental Results

In the experiments below, we first evaluate the performance of DirectBoost on 10 UCI data sets. We then evaluate noise robustness of DirectBoost. For all the algorithms in our comparison, we use decision trees with depth of either 1 or 3 as weak learners since for the small datasets, decision stumps (tree depth of 1) is already strong enough. DirectBoost with decision trees is implemented by a greedy top-down recursive partition algorithm to find the tree but differently from AdaBoost and LPBoost, since DirectBoost does not maintain a distribution over training samples. Instead, for each splitting node, DirectBoost simply chooses the attribute to split on by minimizing 0-1 loss or maximizing the predefined margin value. In all the experiments that $\epsilon$-relaxation is used, the value of $\epsilon$ is 0.01. Note that our empirical study is focused on whether the proposed boosting algorithm is able to effectively improve the accuracy of state-of-the-art boosting algorithms with the same weak learner space $\mathcal{H}$, thus we restrict our comparison to boosting algorithms with the same weak learners, rather than a wide range of classification algorithms, such as SVMs and KNN.

### 3.1 Experiments on UCI data

We first compare DirectBoost with AdaBoost, LogitBoost, soft margin LPBoost and BrownBoost on 10 UCI data sets[1] from the UCI Machine Learning Repository [8]. We partition each UCI dataset into five parts with the same number of samples for five-fold cross validation. In each fold, we use three parts for training, one part for validation, and the remaining part for testing. The validation

| Datasets | N | D | depth | AdaBoost | LogitBoost | LPBoost | BrownBoost | DirectBoost$_{\text{avg}}$ | DirectBoost$^{\epsilon}_{\text{avg}}$ | DirectBoost$_{\text{order}}$ |
|---|---|---|---|---|---|---|---|---|---|---|
| Tic-tac-toe | 958 | 9 | 3 | 1.47(0.7) | 1.47(1.0) | 2.62(0.8) | 3.66(1.3) | **0.63(0.4)** | 1.15(0.8) | 1.05(0.4) |
| Diabetes | 768 | 8 | 3 | 27.71(1.7) | 27.32(1.3) | 26.01(3.3) | 26.67(2.6) | 25.62(2.5) | 25.49(3.0) | **23.4(3.7)** |
| Australian | 690 | 14 | 3 | 14.2(1.8) | 16.23(2.6) | 14.49(4.4) | 13.77(4.6) | 14.06(3.6) | **13.33(3.0)** | 13.48(2.9) |
| Fourclass | 862 | 2 | 3 | 1.86(1.3) | 2.44(1.6) | 3.02(2.3) | 2.33(1.7) | 2.33(1.0) | 1.86(1.3) | **1.74(1.5)** |
| Ionosphere | 351 | 34 | 3 | 9.71(3.7) | 9.71(3.1) | 8.57(2.7) | 10.86(2.8) | **7.71(3.0)** | 8.29(2.7) | **7.71(4.4)** |
| Splice | 1000 | 61 | 3 | 5.3(1.4) | 5.3(2.6) | 4.8(1.4) | 6.1(1.1) | 4.8(0.7) | **4.0(0.5)** | 6.7(1.6) |
| Cancer-wdbc | 569 | 29 | 1 | 4.25(2.5) | 4.42(1.4) | 3.89(1.5) | 4.25(2.2) | 4.96(3.0) | 4.07(2.0) | **3.72(2.9)** |
| Cancer-wpbc | 198 | 32 | 1 | 27.69(7.6) | 30.26(7.3) | 26.15(10.5) | 28.72(8.4) | 27.69(8.1) | **24.62(7.6)** | 27.18(10.0) |
| Heart | 270 | 13 | 1 | 17.41(7.7) | 18.52(5.1) | 19.26(8.1) | 18.15(7.2) | 18.15(5.1) | **16.67(7.5)** | 18.15(7.6) |
| Adult | 6414 | 14 | 3 | 15.6(0.7) | 15.39(0.8) | 16.2(1.1) | 15.56(0.9) | 16.25(1.7) | **15.28(0.8)** | 15.8(1.1) |

Table 1: Percent test errors of AdaBoost, LogitBoost, soft margin LPBoost with column generation, Brown-Boost, and three DirectBoost methods on 10 UCI datasets each with N samples and D attributes.

set is used to choose the optimal model for each algorithm: For AdaBoost and LogitBoost, the validation data is used to perform early stopping since there is no nature stopping criteria for these algorithms. We run the algorithms until convergence where the stopping criterion is that the change of loss is less than $1e$-6, and then choose the ensemble classifier from the round with minimum error on the validation data. For BrownBoost, we select the optimal cutoff parameters by the validation set, which are chosen from {0.0001, 0.001, 0.01, 0.03, 0.05, 0.08, 0.1, 0.14, 0.17, 0.2}. LPBoost maximizes the soft margin subject to linear constraints, its objective is equivalent to DirectBoost with maximizing the average margin of bottom $n'$ samples [19], thus we set the same candidate parameters $n'/n = \{0.01, 0.05, 0.1, 0.2, 0.5, 0.8\}$ for them. For LPBoost, the termination rule we use is same to the one in [6], and we select the optimal regularization parameter by the validation set. For DirectBoost, the algorithm terminates when there is no increment in the targeted margin value, and we select the model with the optimal $n'$ by the validation set.

We use DirectBoost$_{\text{avg}}$ to denote our method that runs Algorithm 1 first and then maximizes the average of bottom $n'$ margins without $\epsilon$-relaxation, DirectBoost$^{\epsilon}_{\text{avg}}$ to denote our method that runs Algorithm 1 first and then maximizes the average margin of bottom $n'$ samples with $\epsilon$-relaxation, and DirectBoost$_{\text{order}}$ to denote our method that runs Algorithm 1 first and then maximizes the bottom $n'$th margin with $\epsilon$-relaxation. The means and standard deviations of test errors are given in Table 1. Clearly DirectBoost$_{\text{avg}}$, DirectBoost$^{\epsilon}_{\text{avg}}$ and DirectBoost$_{\text{order}}$ outperform other boosting algorithms in general, specially DirectBoost$^{\epsilon}_{\text{avg}}$ is better than AdaBoost, LogitBoost, LPBoost and BrownBoost over all data sets except Cancer-wdbc. Among the family of DirectBoost algorithms, DirectBoost$_{\text{avg}}$ wins on two datasets where it searches the optimal margin solution in the region of zero training error, this means that keeping the training error at zero may lead to good performance in some cases. DirectBoost$_{\text{order}}$ wins on three other datasets, but its results are unstable and sensitive to $n'$. With $\epsilon$-relaxation, DirectBoost$^{\epsilon}_{\text{avg}}$ searches the optimal margin solution in the whole parameter space and gives the best performance on the remaining 5 data sets. It is well known that AdaBoost performs well on the datasets with a small test error such as Tic-tac-toe and Fourclass, it is extremely hard for other boosting algorithms to beat AdaBoost. Nevertheless, DirectBoost is still able to give even better results in this case. For example, on Tic-tac-toe data set, the test error becomes 0.63%, more than half the error rate reduction. Our method would be more valuable for those who value prediction accuracy, which might be the case in areas of medical and genetic research.

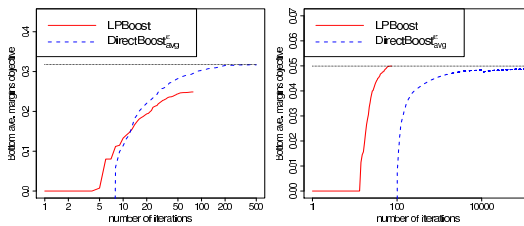

Figure 3: *The value of average margins of bottom $n'$ samples vs. the number of iterations for LPBoost with column generation and DirectBoost$^{\epsilon}_{\text{avg}}$ on Australian dataset, left: Decision tree, right: Decision stump.*

DirectBoost$^{\epsilon}_{\text{avg}}$ and LPBoost are both designed to maximize the average margin over bottom $n'$ samples [19], but as shown by the left figure in Figure 3, DirectBoost$^{\epsilon}_{\text{avg}}$ generates a larger margin value than LPBoost when decision trees with depth greater than 1 are used as weak learners, this may explain why DirectBoost$^{\epsilon}_{\text{avg}}$ outperforms LPBoost. When decision stumps are used as weak learners, LPBoost converges to a global optimal solution, and DirectBoost$^{\epsilon}_{\text{avg}}$ nearly converges to the maximum margin as shown by the right figure in Figure 3, even though no theoretical justification is known for this observed phenomenon.

Table 2 shows the number of iterations and total run times (in seconds) for AdaBoost, LPBoost and DirectBoost$_{\text{avg}}^{\epsilon}$ at the training stage, where we use the Adult dataset with 10000 training samples. All these three algorithms employ decision trees with a depth of 3 as weak learners. The experiments are conducted on a PC with Core2 Duo 2.6GHz CPU and 2G RAM. Clearly

|  | # of iterations | Total running times |
|---|---|---|
| AdaBoost | 117852 | 31168 |
| LPBoost | 286 | 167520 |
| DirectBoost$_{\text{avg}}^{\epsilon}$ | 1737 | 606 |

Table 2: Number of iterations and total run times (in seconds) in training stage on Adult dataset with 10000 training samples and the depth of DecisionTrees is 3.

DirectBoost$_{\text{avg}}^{\epsilon}$ takes less time for the entire training stage since it converges much faster. LPBoost converges in less than three hundred rounds, but as a total corrective algorithm, it has a greater computational cost on each round. To handle large scale data sets in practice, similar to AdaBoost, we can use many tricks. For example, we can partition the data into many parts and use distributed algorithms to select the weak classifier.

## 3.2 Evaluate noise robustness

In the experiments conducted below, we evaluate the noise robustness of each boosting method. First, we run the above algorithms on a synthetic example created by [14]. This is a simple counterexample to show that for a broad class of convex loss functions, no boosting algorithm is provably robust to random label noise, this class includes AdaBoost, LogitBoost, etc. For LPBoost and its variations [25, 26], they do not satisfy the preconditions of the theorem presented by [14], but Glocer [12] showed experimentally that these soft margin boosting methods have the same problem as the AdaBoost and LogitBoost to handle random noise.

| $l$ | $\eta$ | AB | LB | LPB | BB | DB$_{\text{avg}}^{\epsilon}$ | DB$_{\text{order}}$ |
|---|---|---|---|---|---|---|---|
| 5 | 0 | 0 | 0 | 0 | 0 | 0 | 0 |
|  | 0.05 | 17.6 | 0 | 0 | 1.2 | 0 | 0 |
|  | 0.2 | 24.2 | 23.4 | 14.5 | 2.2 | 24.7 | 0 |
| 20 | 0 | 0 | 0 | 0 | 0.6 | 0 | 0 |
|  | 0.05 | 30.0 | 29.6 | 27.0 | 15.0 | 25.4 | 0 |
|  | 0.2 | 29.9 | 30.0 | 29.8 | 19.6 | 29.6 | 3.2 |

Table 3: Percent test errors of AdaBoost (AB), LogitBoost (LB), LPBoost (LPB), BrownBoost (BB), DirectBoost$_{\text{avg}}^{\epsilon}$, and DirectBoost$_{\text{order}}$ on Long and Servedio's example with random noise.

| data | $\eta$ | AB | LB | LPB | BB | DB$_{\text{avg}}^{\epsilon}$ | DB$_{\text{order}}$ |
|---|---|---|---|---|---|---|---|
| wdbc | 0 | 4.3 | 4.4 | 4.0 | 4.5 | 4.1 | 3.7 |
|  | 0.05 | 6.6 | 6.8 | 4.9 | 6.5 | 5.0 | 5.0 |
|  | 0.2 | 8.8 | 8.8 | 7.6 | 8.3 | 8.4 | 6.6 |
| Iono. | 0 | 9.7 | 9.7 | 8.6 | 8.8 | 8.3 | 7.7 |
|  | 0.05 | 10.3 | 12.3 | 9.3 | 11.5 | 9.3 | 8.6 |
|  | 0.2 | 16.6 | 15.0 | 14.6 | 17.9 | 14.4 | 9.5 |

Table 4: Percent test errors of AdaBoost (AB), LogitBoost (LB), LPBoost (LPB), BrownBoost (BB), DirectBoost$_{\text{avg}}^{\epsilon}$, and DirectBoost$_{\text{order}}$ on two UCI datasets with random noise.

We repeat the synthetic learning problem with binary-valued weak classifiers that is described in [14]. We set the number of training examples to 1000 and the labels are corrupted with a noise rate $\eta$ at 0%, 5%, and 20% respectively. Examples in this setting are binary vectors of length $2l+11$. Table 3 reports the error rates on a clean test data set with size 5000, that is, the labels of test data are uncorrupted, and a same size clean data is generated as validation data. AdaBoost performs very poor on this problem. This result is not surprising at all since [14] designed this example on purpose to explain the inadequacy of convex optimization methods. LogitBoost, LPBoost with column generation, and DirectBoost$_{\text{avg}}^{\epsilon}$ perform better in the case that $l = 5$ and $\eta = 5\%$, but for the other cases they do as bad as AdaBoost. BrownBoost is designed for noise tolerance, and it does well in the case of $l = 5$, but it also cannot handle the case of $l = 20$ and $\eta > 0\%$. On the other hand, DirectBoost$_{\text{order}}$ performs very well for all cases, showing DirectBoost$_{\text{order}}$'s impressive noise tolerance property since the most difficult examples are given up without any penalty.

These algorithms are also tested on two UCI datasets, randomly corrupted with additional label noise on training data at rates of 5% and 20% respectively. Again, we keep the validation and the test data are clean. The results are reported in Table 4 by five-fold cross validation, the same as Experiment 1. LPBoost with column generation, DirectBoost$_{\text{avg}}^{\epsilon}$ and DirectBoost$_{\text{order}}$ do well in the case of $\eta = 5\%$, and their performance is better than AdaBoost, LogitBoost, and BrownBoost. For the case of $\eta = 20\%$, all the algorithms perform much worse than the corresponding noise-free case, except DirectBoost$_{\text{order}}$ which still generates a good performance close to the noise-free case.

## 4 Acknowledgements

This research is supported in part by AFOSR under grant FA9550-10-1-0335, NSF under grant IIS:RI-small 1218863, DoD under grant FA2386-13-1-3023, and a Google research award.

## Footnotes

[1]For Adult data, where we use a subset a5a in LIBSVM set http://www.csie.ntu.edu.tw/~cjlin/libsvm. We do not use the original Adult data which has 48842 examples since LPBoost runs very slow on it.

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
