[Reviews · NeurIPS 2013]

Submitted by Assigned_Reviewer_4

The paper studies the idea of using a boosting-like approach to directly minimize the training error and various functions of the training margins. The algorithms are explained in detail, and decent experimental results are presented. Theory is fairly minimal.

In a way, this is a fairly obvious idea, but conventional wisdom says that the idea should not work since classification error is not convex or even continuous. It is great to see someone try it, and to spell out all the issues involved, and the details of how to implement it. I thought the experimental results were especially strong, a nice comparison on 10 datasets against several boosting algorithms. The new method works surprisingly well, especially in the presence of noise.

The presentation is generally good. I thought the descriptions of algorithms could have been a bit more precise, but the paper gives examples that really help to illustrate what is going on.
Summary: A good idea presented with algorithmic details, and strong experimental results.

Submitted by Assigned_Reviewer_7

In this paper the authors provide an algorithm for directly minimzing 0-1 loss and margin maximization. Most existing machine learning techniques have relied on minimizing a convex upper bound on the 0-1 loss in classification problems. In contrast, in this paper the authors propose a simple greedy algorithm for directly minimizing the 0-1 loss via a combination of weak learners. This is followed by a few steps of direct maximization of margin. The proposed algorithm is then evaluated on a few small low dimensional datasets.

I have the following major concerns about this paper:

The authors claim that their algorithm has a much favorable run-time compared to AdaBoost (Table 2). I do not understand how this is possible. In AdaBoost, in each round, a weight is given to a training example and a weak learner such as a decision tree can be found quite efficiently to minimize a weighted loss (for example using CART as the weak learner). Thus, in each step of boosting one needs to find only one decision tree. However, for the proposed algorithm, one has to iterate over all possible weak learners. I just don't see how the proposed algorithm can be computationally more efficient unless the number of weak learners is really small. In many applications with large datasets it is typical to consider decision trees of depth 5 or 10 and I do not see how the proposed method can be efficient in that case if one has to enumerate all weak learners. I am concerned that the proposed algorithm is limited to small datasets with low dimensions and weak learners with very few instances.

From the standard deviations in Table 1, I am not sure that the proposed method results in statistically significant results in Table 1 compared to AdaBoost. If we take the standard error most of the results seem to overlap.

The proposed method consists of two steps: first, 0-1 loss is minimized using greedy co-ordinate descent. Once a few weak learners are selected, a few more weak learners are added to maximize the average margin of bottom n' examples greedily. How much of the claimed benefit is due to each step? For example, what happens if you run your algorithm 2 after a few iterations of AdaBoost? "Direct 0-1 loss minimization" is a bit of a misnomer since it is followed up with a few steps of margin maximization. I am not sure such natural questions are adequately answered in this paper.

Summary: While this work could be potentially interesting, I am not completely convinced about this paper at this point.

Submitted by Assigned_Reviewer_8

This paper presents a boosting method that directly minimizes the empirical classification error that is defined based on the indicator whether predicted y = observed y, the so-called 0-1 loss. The proposed method first runs greedy coordinate descent to search the coordinatewise minimum loss, and then runs coordinate ascent to expand the margin. The method is interesting and novel, and offers some advantages. The following concerns are raised.

1. It seems the authors assume a hypothesis space which has a finite number of hypotheses because in the algorithms 1 and 2, it loops on all hypotheses h at each iteration. For instance, algorithm 1 finds all weak learners that lead to largest classification error reduction at each iteration. What if users choose a hypothesis space, such as linear functions?

2. At the end of Algorithm 1, what is the rationale to update the weight of one weak learner that gives the smallest exponential loss? Isn’t it still using a convex loss function although in the early part of the algorithm, it uses 0-1 loss.

3. It also assumes that data is always separable as long as the combined strong classifier is strong. What if it is not the case? What if the chosen hypothesis space is not complex enough to separate a given dataset. The paper states that algorithm 1 reaches a solution of coordinatewise minimum. It also says the 0-1 loss reaches 0. What is the definition of coordinatewise minimum? Is it actually a global minimum because the lowest error would be 0.

4. When the second step (margin maximization) starts, it starts from a region that would not get the solution out of the 0 loss region, characterized by the value of d. It is not clear how exactly this value of d is calculated instead of simply saying “determine the lowest sample whose margin is decreasing”.

5. Given separable cases are assumed, the margin is always positive in their formulation. What happens if there are negative margins for inseparable cases? The second part of the algorithm would not run.

6. In the early paragraph of page 2, it says “it might escape the region of minimum training error in order to achieve a larger margin”. In the later section, the design of algorithm 2 will not allow the margin maximization step go beyond zero 0-1 loss region.

7. Assuming a weak learner can be obtained in a similar computational cost, DirectBoost is certainly more computationally heavy for each iteration than AdaBoost. In their experiments, what is the stopping criterion that they used for AdaBoost? Why does AdaBoost need so many iterations?

8. It is not that clear about the truly convincing advantage of the proposed method over regular boosting methods.
Summary: This paper presents a boosting method that directly minimizes the empirical classification error that is defined based on the indicator whether predicted y = observed y, the so-called 0-1 loss. The method is interesting and novel, and offers some advantages although there are some concerns.

Submitted by Assigned_Reviewer_13

Simple idea with nice results.
If this is effectively new, then the paper should be accepted.
Therefore the novelty aspect should be carefully checked.

Is it really a boosting algorithm? To me it should be called a pursuit algorithm because it does not apparently rely on manipulating the distribution of examples. This is important because there are closely related works (see kernel matching pursuit, http://www.iro.umontreal.ca/~vincentp/Publications/kmp_techreport2000.pdf, section 3) found in the pursuit literature instead of the boosting literature. This is close, but not exactly the same, though...

About the overlap with papers 539 and 956. I was initially a reviewer of paper 956 which describes a multiclass variant of the same idea. Paper 956 is harder to understand and could have been a paragraph in paper 481. Paper 539 is a semi-supervised variant od the same idea and could have been a paragraph in paper 481. This splitting of a work into minimal publishable units is unwise. I think that the community will be better served by accepting 481, rejecting 539 and 956, and letting the author know that he can refer to these extensions in the final version of the Nips paper and defer the full discussion to the future journal paper.


Detailed comments:

[page 1, line 44] -- Even with surrogate losses, the problem is not always convex. This depends on the family of functions.
Summary: Recommend accepting only 481 which contains the important idea.
Author Feedback

Author rebuttal: For Reviewer 13:
Thank you for your review. You are correct that our method doesn’t maintain a distribution of samples, so it is similar to a matching pursuit algorithm in this sense. However, according to Hastie's book [1], page 605, "... Boosting was initially proposed as a committee method as well, although unlike random forests, the committee of weak learners evolves over time, and the members cast a weighted vote.” Clearly our method is a boosting method in this sense. We haven't get a chance to show our method is a boosting algorithm in the PAC learning sense, but the term "boosting" has come to be used for a range of algorithms, and a lot boosting algorithms do not possess the PAC-boosting property. 
Thank you for your comments about related papers 956 and 539. You are right that surrogate functions are not always convex.

For Reviewer 4:
Thank you for the valuable feedback. We do have theoretical results of the generalization error bound in terms of the average margin of bottom n′ samples or n′th order margin of training examples. So we add a statement "Instead, we can prove that the generalization error of any ensemble classifier is bounded in terms of the average margin of bottom n′ samples or n′th order margin of training examples, as well as the number of training examples and the complexity of the base classifiers."

For Reviewer 7
Thank you for your comments and concerns. Due to space limitations, we were only able to give very brief descriptions of building trees in lines 309-313 in the paper. Similar to other boosting methods, DirectBoost uses a greedy top-down partition strategy in each step to recursively find a single tree; it does not need to enumerate all trees. For each splitting node, our algorithm looks for a split that leads to better 0-1 loss or margin objective. When we use binary split, our algorithm has the same computational complexity as AdaBoost with CART as a weak learning algorithm. When we use multiway split, our algorithm has higher computational complexity, but as pointed out by Hastie, page 311 in [1], binary split is preferred nonetheless. Therefore we used binary split for all boosting algorithms in our experiments. As can be seen in Table 2, for each iteration, DirectBoost is slightly slower than AdaBoost.
On lines 340 and 346, we describe the stopping criterion for AdaBoost and DirectBoost, respectively. Table 2 shows the running times. Though DirectBoost is slower in each iteration, it requires much fewer iterations and is therefore more efficient than AdaBoost overall.
In Table 1, comparing DirectBoost^epsilon_avg to AdaBoost, many of the results are overlapped if we consider the standard error across the folds. However, DirectBoost_avg^epsilon gives better results, on average, in 9 out of 10 data sets, which shows the effectiveness of our methods.
Algorithm 1’s objective is to minimize 0-1 loss, and algorithm 2 looks for a model with an optimal margin objective under the condition of minimum 0-1 loss in a certain sense. While algorithm 1 can be substituted with AdaBoost, the problem of AdaBoost is that when the data is not linearly separable, it does not have a good stopping criterion. Algorithm 1 stops at a coordinatewise local minimum.
“Direct” in our title is an adjective not only for “0-1 loss minimization” but also for “margin maximization.”

For Reviewer_8
Thank you for your comments and questions. In this paper, we only consider the cases, such as decision trees, where the hypothesis space is finite. We will consider how to extend to the case where the hypothesis space is infinite as future work.
When there are several weak learners with the same 0-1 loss, we choose the one with the smallest exponential loss; when we identify an interval with the same 0-1 loss, we can choose any point in the interval, such as the midpoint. However, we found empirically that choosing the point with the minimum exponential loss in the interval leads to better results.
The definition of coordinatewise local minimum can be found on page 479 in [2]. When the data is not separable, algorithm 1 reaches a coordinatewise local minimum, then Algorithm 2 searches a model with a larger margin objectives. In this case, there are some samples, each having a negative sample margin. Suppose the 0-1 loss is 5%, then Algorithm 2 starts here and interval [0,d) is the one with training error<=5%.
We use epsilon-relaxation to escape the region of zero (or local) 0-1 loss. This leads to a value of alpha_t that is epsilon greater than d.
For each iteration, DirectBoost is slightly slower than AdaBoost. On lines 340 and 346, we describe the stopping criterion for AdaBoost and DirectBoost, respectively. Table 2 shows the running times. Though DirectBoost is slower in each iteration, it requires much fewer iterations and is therefore more efficient than AdaBoost overall.
One of the advantages of our boosting method might be its robustness to label noise, as described in Section 3.2.

[1] Hastie etal. The Elements of Statistical Learning: Data Mining, Inference, and Prediction, 2nd Edition, 2009.
[2] P. Tseng. Convergence of block coordinate descent method for nondifferentiable minimization. Journal of Optimization Theory and Applications, 475--494, 2001.